# Tackling the Future Pandemics: Broad-Spectrum Antiviral Agents (BSAAs) Based on A-Type Proanthocyanidins

**DOI:** 10.3390/molecules27238353

**Published:** 2022-11-30

**Authors:** Massimo E. Maffei, Cristiano Salata, Giorgio Gribaudo

**Affiliations:** 1Department of Life Sciences and Systems Biology, University of Turin, Via Quarello 15/a, 10135 Turin, Italy; 2Department of Molecular Medicine, University of Padua, 35121 Padua, Italy; 3Department of Life Sciences and Systems Biology, University of Torino, Via Accademia Albertina 13, 10123 Turin, Italy

**Keywords:** polyphenols, proanthocyanidins, A-type linkage, human viruses, broad-spectrum antiviral activity

## Abstract

A-type proanthocyanidins (PAC-As) are plant-derived natural polyphenols that occur as oligomers or polymers of flavan-3-ol monomers, such as (+)-catechin and (−)-epicatechin, connected through an unusual double A linkage. PAC-As are present in leaves, seeds, flowers, bark, and fruits of many plants, and are thought to exert protective natural roles against microbial pathogens, insects, and herbivores. Consequently, when tested in isolation, PAC-As have shown several biological effects, through antioxidant, antibacterial, immunomodulatory, and antiviral activities. PAC-As have been observed in fact to inhibit replication of many different human viruses, and both enveloped and non-enveloped DNA and RNA viruses proved sensible to their inhibitory effect. Mechanistic studies revealed that PAC-As cause reduction of infectivity of viral particles they come in contact with, as a result of their propensity to interact with virion surface capsid proteins or envelope glycoproteins essential for viral attachment and entry. As viral infections and new virus outbreaks are a major public health concern, development of effective Broad-Spectrum Antiviral Agents (BSAAs) that can be rapidly deployable even against future emerging viruses is an urgent priority. This review summarizes the antiviral activities and mechanism of action of PAC-As, and their potential to be deployed as BSAAs against present and future viral infections.

## 1. Introduction

One of the most unsettling lessons that the Coronavirus disease 2019 (COVID-19) pandemic has taught the world is its general unpreparedness for tackling a new respiratory virus pandemic by a therapeutic approach [1]. Notwithstanding that the Severe Acute Respiratory Syndrome Corona Virus 1 (SARS-CoV-1) (2003) [2], and the Middle East Respiratory Syndrome Coronavirus (MERS-CoV) (2012) [3] outbreaks proved the risk of the emergence of new zoonotic coronaviruses, the lack of already available and effective Broad-Spectrum Antiviral Agents (BSAAs), rapidly deployable against the new SARS-CoV-2, made it difficult in the beginning to reduce hospitalizations and deaths, as well as to slow down the spread of COVID-19 [4]. Thus, new BSAAs that can be rapidly deployed against future emerging respiratory viruses in humans, such as coronavirus and influenza virus, are urgently needed. Such BSAAs might allow antiviral treatments to begin immediately after the virus emergence, thus gaining time for the development of the new virus specific vaccines and therapeutics [5].

BSAAs are compounds that inhibit the replication of a wide range of viruses, since different viruses share similar biochemical pathways to synthetize their components and/or exploit the same cellular molecules and pathways to replicate in the host [6]. Given the substantial diversity in viral structures and replication strategies, the development of effective BSAAs has proven to be more difficult than that of the most approved antiviral drugs (i.e., Direct-Acting Antivirals, DAAs), that inhibit only a specific virus-encoded target, such as a polymerase or a protease. Nonetheless, in the last two decades the increasing number of new viral outbreaks in humans has restated the critical need for molecules able to implement the “one drug, multiple virus” paradigm, that is the inhibition of viruses from different families by the same molecule. Thus, effective BSAAs could constitute an essential weapon in the ultimate arsenal of available antiviral options, as they could provide immediate therapeutic intervention against emerging and re-emerging viral threats.

Based on the target, BSAAs can then be categorized into two main types: (1) compounds that target viral structures or enzyme activities, and thus belong to the more general category of DAAs; and (2) compounds that affect host factors or cellular biochemical pathways essential for viral replication, and therefore defined as host-targeted antivirals (HTAs) [7].

The main advantage of BSAAs over the most current approved virus-specific DAAs that are likely inadequate for treating new emerging viruses, consists in their activity not only against viruses belonging to different families, but also towards different genotypes of the same virus species; therefore, they are potentially effective also against viruses not yet emerged in humans. Because of this, BSAAs are suitable as first-line treatments for emerging respiratory virus outbreaks or new sexually transmitted infections, thanks to their rapid repositioning from one pandemic event to the next emerging one. Moreover, host-targeted BSAAs have the inherent edge of a high barrier to the emergence and development of viral drug resistance [6,7]. However, HTAs are burdened with the potential high risk of cellular toxicity, as well as a poor in vitro-to-in vivo translation caused by the systemic compensation of the effects of the blockage of a specific cellular pathway [6]. On the other hand, BSAAs that directly target virus components, such as some DAA nucleoside analogues, although endowed with a lesser potential for host cell toxicity compared to host-targeted BSAA, are prone to the selection of drug-resistant strains [7]. Obviously, the balancing, at least theoretically, of these disadvantages could be accepted in the design of new BSAA-based intervention strategies depending on the threat posed by an emerging viral infection, the characteristics of the causative agent, and the length of treatment, with the final goal of increasing the BSAA’s therapeutic window [6].

Because the development of effective BSAAs remains a challenging task in drug discovery, natural products have been considered as a unique source of chemical complexity and diversity within which antiviral activities can be identified. Indeed, an increasing body of evidence, based on robust molecular, biochemical and pharmacological studies, indicates that a wide-range of natural products derived from plants show inhibitory effects on the replication of many different viruses, thus having the potential to be deployed as BSAAs against both current viruses and new emerging viral threats [8,9,10]. The bioactive components of some of these plant extracts endowed with antiviral activity have been identified as specific polyphenols, flavonoids, glucosides, terpenes, and alkaloids, and the mechanisms of action and molecular targets for some of these molecules have been elucidated [11,12]. Among the large category of natural polyphenols, proanthocyanidins (PACs) characterized by unusual double A linkages of the component catechins monomers (A-type PACs, PAC-As), have been reported to recapitulate the antibacterial and antiviral effects of several plant extracts from which they were isolated and characterized [13,14,15].

This review focuses on A-type PACs as the bioactive chemical components of plant extracts with antiviral activities, with emphasis on their structure, natural origin, mechanism of the antiviral actions including the molecular targets, and their potential to be developed as BSAAs.

## 2. A-Type Proanthocyanidins’ Chemistry and Where They Can Be Found

It is known that plants biosynthesize bioactive molecules to fulfil their physiological needs, such as defense against herbivores and pathogens, as well as for the interspecific allelopathic competition with other plants [16]. Among these molecules, PACs are interesting bioactive polyphenols that derive from at least two or more 2-phenyl-3,4-dihydro-2H-chromen-3-ol (flavan-3-ol) units which can be condensed through a single (B-type) or a double (A-type) bond [17] (Figure 1). Flavan-3-ols have a saturated A-ring which makes PACs non-planar molecules [18]. PACs are quite complex molecules with a variety of structures because of the stereochemistry of flavanol heterocycle, the type of linkage among the different units, and more importantly, the number and position of the hydroxyl groups linked to the aromatic rings [19]. C–O bonds between the oxygen in position 7 (O_7_) of one flavan-3-ol unit and the carbon in position 2 (C_2_) of another unit generate A-type PACs. Because the hydroxyl group linked to the C-ring of each flavan-3-ol can be in either *S* or in *R* configuration, different typologies of A-type PACs can be formed.

The biosynthesis, transport and polymerization of PACs, as well as the synthetic efforts made to obtain both naturally occurring A-type PACs and their structurally simplified analogues have been recently reviewed [19].

The distribution of natural PACs in plants is quite large and many red fruits contain different amounts of these bioactive molecules. However, one of the major problems in PAC quantification in plant extracts is the variability of used methods (e.g., the classical gravimetric methods, colorimetric methods based on acid butanol, the ethanol/butanol method, the vanillin assay, and the Brunswick Laboratories 4-dimethylaminocinnamaldehyde or BL-DMAC assay), which represents a bottleneck in the classification of natural sources containing these active polyphenols. From an analytical point of view, HPLC coupled to mass spectrometry, the Matrix-Assisted Laser Desorption/Ionization (MALDI), and ion-mobility mass spectrometry are the best methods for the characterization of both the type of bonds and degree of polymerization [20,21,22,23]. The BL-DMAC method is actually the most widely used method, and although originally developed for the quantification of PACs in cranberry extracts, it proved to be a reliable method also for quantification of PAC in different plant sources [24,25,26,27,28,29,30].

Restricting the natural sources of PACs to those characterized through the BL-DMAC method, Mannino and co-workers [19] performed a meta-analysis that made it possible to identify plant species belonging to 35 different families. The results showed that PACs occur primarily in fruits and seeds, and less in fruit skins (with the exception of peanut skins), leaves and resins. As expected, the meta-analysis showed that the most represented plant family is the Ericaceae, and especially the genus *Vaccinium* with 10 different species characterized by high contents of PACs. Other high-PAC species such as *Styrax ramirezii* (fam. Styracaceae) and *Carya illinoinensis* (fam. Juglandaceae) were also identified. Owing to the biological activities of A-Type PACs, we extended the search for plant sources that mostly contain this type of PACs, and listed their source, biological activity, and chemical properties in Table 1 below.

A-type PACs from different natural sources have been observed to exert different biological effects, such as antiviral (see below), antioxidant, antibacterial, and immunomodulatory activities. Based on the data summarized in Table 1 and Table 2 (see below), we performed a Principal Component Analysis (PCA) that correlates the presence (1) or the absence (0) of different PAC-A degrees of polymerization with three main biological activity (antiviral, antibacterial and antioxidant) by using a single linkage method, with Pearson distances and a varimax rotation. The results of this PCA analysis that are depicted in Figure 2 show that the antiviral activity of PAC-As is correlated primarily to the presence of low-degrees of A-type PAC polymerization (from monomers to tetramers), whereas the antibacterial activity (primarily against bacteria that cause Urinary Tract Infections, UTI) was associated with the presence of dimeric and trimeric PAC-A. Most of the data summarized in Table 1 show that plants that possess polymeric A-type PACs (from pentamers to dodecamers) display antioxidant activity. This observation is confirmed by our PCA analysis (Figure 2).

## 3. The Broad-Spectrum Antiviral Activity of A-Type PACs

In this section we focus on the activity of PACs as BSAAs, with particular reference to A-type PACs. The studies that have identified A-type PACs as specific components responsible for the antiviral activities of several natural extracts are summarized below, taking into consideration the human viruses for which the inhibitory activity has been characterized. PAC-As with antiviral activity are listed in Table 2 and described in the following paragraphs.

### 3.1. Herpes Simplex Virus

Herpes Simplex Virus type 1 (HSV-1) and type 2 (HSV-2) cause lifelong infections with periodic reactivations that are highly prevalent worldwide [198,199]. A wide range of diseases result from HSV infections, from the most common cold sores and genital herpes, to recurrent keratitis, and even life-threatening systemic infections and encephalitis [198,199]. Antiviral intervention is therefore needed for the therapy of these diseases. However, the currently available DAAs cannot eliminate an established latent infection, and their prolonged administration may lead to the occurrence of viral resistant strains as well as toxicity [200]. Therefore, the development of new anti-HSV agents that may even prevent the establishment of an HSV infection is a significant medical need [200].

In this regard, many different small molecules from plant extracts, such as polyphenols, terpenes, and flavonoids have been described as exerting an anti-HSV activity in vitro [201,202,203]. Among polyphenols, PACs have been identified as the bioactive anti-HSV agents through chemical and biological characterization of fractions derived from several plant extracts [175,204,205,206,207,208]. In some studies, the antiviral activity of PACs against both HSV-1 and HSV-2, was observed to stem from the ability of PACs to inhibit the virus attachment to the cell surface and the subsequent entry into host cells [209,210]. Especially, Gescher et al. [204] observed that the epicatechin-3-*O*-gallate-(4→8)-epicatechin-3-*O*-gallate, a dimeric B-type PAC, isolated from an acetone-water extract obtained from the aerial parts of *Rumex acetosa*, interacted directly with purified HSV-1 particles and provoked the oligomerization of gD, an essential envelope glycoprotein required for the virus binding to cellular receptors [211]. It was concluded that the *R. acetosa*-derived PAC-B2 inhibited HSV-1 replication as a result of its ability to bind infectious viral particles and tampering with gD, thus preventing efficient interactions with cell surface receptors [204].

As regards the anti-HSV activity of A-type PACs, in an early study, a series of PAC dimers was isolated from an extract of a byproduct in cocoa production, and then tested for antiviral activity by De Bruyne et al. [124]. Among the different PAC dimers examined, the PAC-A1 or epicatechin-(4β→8, 2β→*O*→7)-catechin, and the PAC-A2 or epicatechin-(4β→8, 2β→*O*→7)-epicatechin, were observed to exert the most potent inhibitory activity against in vitro replication of HSV, inasmuch a 4-log reduction in viral titer was measured in the presence of 100 µg/mL of either PAC-A1 or PAC-A2 compared to untreated controls (16). Later, Xu et al., [181] isolated seven A-type PACs from an alcoholic extract of lychee (*Litchi chinensis*) seeds, that were then examined for antioxidants and antiviral activities. In this study, an anti-HSV-1 activity of a lychee-derived PAC-A2 was determined in Vero cells, with an EC_50_ of 18.9 µg/mL, and a Specificity Index (SI) of 3.0 [181].

In a subsequent study, oligomeric A-type PACs fractionated from an extract of *Chamaecrista nictitans* were observed to be related to the overall anti-HSV activity of the extract [175].

More recently, we examined the suitability of a cranberry extract as a direct-acting anti-HSV agent [188]. Analysis of the anti-HSV activity of purified fractions revealed that the ability of the whole cranberry extract to hinder HSV replication was due to its high content of type-A PACs. In fact, only the fraction that contained PAC-A dimers and small amounts of trimers exerted an antiviral activity against HSV-1 and HSV-2 replication in Vero cells, with EC_50_ of 19.2 and 6.8 µg/mL and SI of 9.5 and 27.6, respectively [188]. Then, mechanistic investigations highlighted that the whole extract or its PACs-A-containing fraction interacted with the envelope glycoproteins gD and gB, the fusion protein of the HSV machinery for entry that carries out membrane fusion [211], thus causing a loss of infectivity of HSV particles [188].

It is therefore possible to recapitulate a common mechanism of action of the anti-HSV activity of both PAC-A and -B [188,209] that could depend from their ability to interact with viral envelope glycoproteins. These interactions, in turn, may affect the functions of those glycoproteins required for HSV attachment and entry, such as gD and gB, thus preventing these initial phases of the HSV replication cycle.

### 3.2. Human Immunodeficiency Virus (HIV)

Acquired Immunodeficiency Syndrome (AIDS), caused by the Human Immunodeficiency Virus 1 (HIV-1), is an immunosuppressive disease that creates susceptibility to lethal opportunistic infections and malignancies [212]. Although many drugs have been approved and increase the quality of the life of infected people, the high costs and the life-long treatments makes therapy a hard goal in low-income countries. In addition, viral drug resistance prompts researchers to develop new antiviral agents. In this context, several Authors investigated plant extracts to identify anti-HIV-1 activities [213]. Among bioactive compounds able to interfere with HIV-1 infection, PAC-As have been identified as candidates for new antivirals development. In 1999, De Bruyne and co-workers [124] evaluated the biological effects and antiviral activity of PAC-As and related polyphenols. They reported that PAC-A1 and PAC-A2 were the most potent antiviral compounds, reducing the HIV-1 cytopathic effects (CPE) in infected cells with EC_50_ of 14 and 5.8 µg/mL and SI of 10 and 24, respectively [124]. The anti-HIV-1 mechanism of PAC-As was then elucidated by Fink and coworkers [176]. They observed that elderberry and cinnamon extracts incubated with the virus during the infection step significantly reduced the number of foci of infected cells, with an EC_50_ from 0.5 to 201 µg/mL for four different HIV-1 types. A direct binding assay coupled with a mass spectrometry approach then showed that PAC-As interacted with viral particles, thereby reducing the virus infectivity. The interaction of PAC-As with HIV-1 particles followed a stoichiometric pattern, thus suggesting HIV-1 envelope glycoproteins as the specific viral target [176]. Furthermore, PAC-As also showed a synergistic effect with the antiretroviral drug enfuvirtide, a drug interacting with the envelope gp41 subunit that blocks the fusion of the HIV-1 to target cells. Indeed, the PAC-As-mediated antiviral activity, being not competitive with enfuvirtide, was suggested as being most likely to target the gp120 subunit [176].

More recently, the anti-HIV-1 activity of the cinnamon-derived compound IND02, that contains A-type PAC trimers and pentamers, was reported [177]. Using surface plasmon resonance, the authors showed that IND02 and IND02-trimer bind to gp120 of HIV-1 types that use CXCR4 (X4, lympho-tropic strain) or CCR5 (R5, macrophage-tropic strain) as co-receptors [177]. Because HIV-1 infection requires multiple interactions of the gp120 with host molecules, such as heparansulfate (HS), the viral receptor CD4, and the R5/X4 co-receptors, the potential of IND02 to interfere with different interaction stages of HIV-1 attachment and entry was investigated. IND02 and IND02 trimer were observed to inhibit the gp120-HS binding in a concentration-dependent manner, while only IND02 affected the gp120-CD4 interaction, as well as the binding of gp120 of R5- and X4-tropic viruses. These results suggested that IND02 could interact with the gp120’s CD4 binding domain of both R5 and X4-tropic viruses, probably to the protein domains involved in interactions with co-receptors. Finally, the antiviral activity of IND02 was confirmed in a biological assay by its addition during the infection step of activated human peripheral blood mononuclear cells (PBMCs) with a panel of clinically relevant primary strains, for which low micromolar EC_50_ values were observed [177].

Of note, Suedee and colleagues reported a new anti-HIV-1 mechanism PAC-As. With the aim to investigate the anti-integrase (IN) activity of some Thai medicinal plant extracts, they discovered that PAC-A_2_ derived from a leaf extract of *Pometia pinnata* inhibited the HIV-1 enzyme with an IC_50_ value of 30.1 µM. However, this result was obtained from an in vitro enzymatic assay, and no evidence of this PAC-A_2_ activity in the context of HIV-1 infection was reported [184]. Moreover, Tietjen et al. [180] identified the ixoratannin A-2 as HIV-1 inhibitor with an EC_50_ value of 35 µM. Ixoratannin A-2 is a doubly linked A-type PAC trimer isolated from the *Ixora coccinea* shrub collected in western Nigeria, and it was suggested that ixoratannin A-2 might inhibit the ion channel activity of the viral protein Vpu [180]. More recently, and relevant to this hypothesis, a computational study indicated that ixoratannin A-2 might interact with several human and viral proteins, included Vpu [214], thus supporting Tietjen et al.’s hypothesis [180].

Additional studies reported that plant-derived procyanidins other than PAC-As can affect HIV-1 infection. For example, Nair and co-workers [197], reported that grape seed extract-derived PACs inhibited HIV-1 infection by downregulating the co-receptors on the surface of PBMCs [197]. In addition, Feng et al. [182] reported that a procyanidin-rich extract from French maritime pine not only affected HIV-1 virus entry but also its genome replication. Because the prominent biochemical alteration induced in target cells by the French maritime pine consists of an overexpression of the Mn-superoxide dismutase, an intracellular antioxidant protein, its involvement was suggested in the overall anti-HIV-1 activity [182]. Thus, the modulation of some stress-induced cellular pathways by PACs may represent an additional strategy to counteract HIV-1 infections.

### 3.3. Chronic Hepatitis Viruses

Hepatitis B virus (HBV) and Hepatitis C virus (HCV) are a major cause of liver disease worldwide. HBV is an enveloped double stranded DNA virus (*Hepadnaviridae*) while HCV is a enveloped positive-strand RNA virus (*Flaviviridae*), both characterized by hepatic tropism. Transmission happens through intra-family contacts among infants, by sexual or parenteral contact or by the vertical route. For both viruses, after a possible acute phase, viral infection may progress in chronicity. During chronic infection, viral cytopathic effects combined with the cell damage due to the immune response may promote liver cirrhosis and hepatocellular carcinoma [215]. Although effective therapies for the treatment of HBV and HCV infection are available, with a clear improvement of patient treatments and the cure of the infection, at least for HCV, the high costs of the therapy and the risk of drug failure still prompt to the search and development of new drugs.

Many efforts have been addressed to the identification of natural products as cheaper and more accessible sources of new anti-HBV agents [216]. With the aim of discovering potential anti-HBV molecules, Tsukuda and co-workers [217] identified PACs as HBV inhibitors. PACs inhibited HBV infection both in cell lines and in primary human hepatocytes by blocking viral particles’ attachment to target cells (EC_50_ of ~8 μM) without any effect on viral genome replication and cell viability. Using biochemical assays, it was observed that PACs interacted with the preS1 region of the viral glycoprotein. In addition, PACs showed an anti-HBV effect against multiple viral genotypes and one viral isolate resistant to the approved antiviral drug entecavir. In contrast to other known molecules that interfere with the HBV life cycle, the antiviral activity of PACs directly targets the viral particle, thus acting as a virucidal agent.

Concerning HCV, Takeshika et al. [196,218] reported that purified PACs (PAC-B primarily) from blueberry leaves inhibited HCV RNA replication (EC_50_ 0.087 µg/mL, SI 212). This antiviral activity was evaluated using an HCV subgenomic expression system, while the adhesion/internalization stages of viral particles were not investigated. However, it was observed that blueberry leaf-derived PACs interacted with the heterogeneous nuclear ribonucleoprotein A2/B1 that is indispensable for HCV subgenome expression. Moreover, the anti-HCV activity was found dependent on the polymerization level of PACs, reaching the maximum efficacy with a polymerization degree between 8 and 9 [196,218]. Similarly, Li and coworkers [219] reported that PAC-B1 purified from a cinnamon bark extract inhibited HCV RNA synthesis in a concentration-dependent manner in Huh-7 cells, but it did not interfere with viral entry or receptor expression [219]. As for HIV-1, a French maritime pine extract was reported to inhibit HCV. Since oxidative stress has been identified as a key mechanism of HCV-induced pathogenesis, Ezzikouri et al. [183] evaluated the antiviral properties of a French maritime pine extract in both in vitro and in vivo models. Using HCV replicon cell lines, the authors reported both the inhibition of the HCV replication (EC_50_ ~ 40 μg/mL) and the reduction of ROS [183]. In addition, treatment of infected chimeric mice with the same extract suppressed HCV replication and showed a synergistic effect with interferon-alpha [183].

Regarding PAC-A, the cinnamon-derived compound IND02 was tested in Huh7.5.1 cells and primary human hepatocytes (PHH) by using HCV and HCV pseudoparticles [178]. IND02, added to target cells for one hour before viral infection, showed a concentration-dependent inhibitory effect against both the wild type virus and a difficult-to-treat HCV strain, characterized by enhanced cell entry efficiency and poor neutralization by neutralizing antibodies [178]. To shed light on the stage of the HCV replication cycle affected by IND02, the authors used HCV pseudoparticles and a subgenomic replicon system to investigate IND02 activity on virus entry and replication, respectively. The results showed that IND02 markedly inhibited the first stage of infection in a way that overlapped with the inhibitory activity of the anti-CD81 antibody that targets the HCV cell entry factor CD81. Then, a kinetics experiment showed that IND02 inhibited HCV infection when added after HCV attachment to target cells, thus suggesting an interference with the internalization of adsorbed virus particles or with the membrane fusion step [178].

Finally, the anti-HIV-1 ixoratannin A-2, a PAC-A trimer from the *Ixora coccinea* [180], was observed to also inhibit HCV replication in Huh-7 cells with an EC_50_ of 23.0 μM.

Interestingly, PAC-A and PAC-B seem to target different stages of the HCV replication cycle by acting mainly at the level of viral entry or viral RNA replication, respectively.

### 3.4. Enteric Viruses

Enteric viruses are a major cause of morbidity and mortality, especially among children in developing countries [220]. Different families of human viruses include agents that target the gastrointestinal tract to cause gastroenteritis, diarrhea, and hepatitis [220]. Members of the *Picornaviridae* (e.g., enterovirus, hepatitis A virus, Aichi virus), *Reoviridae* (e.g., rotavirus), *Caliciviridae* (e.g., norovirus), *Astroviridae* (e.g., astrovirus), *Hepeviridae* (e.g., hepatitis E virus) and *Adenoviridae* (e.g., adenovirus 40 and 41) are in fact major enteric viral pathogens [220,221,222]. These viruses represent a major public health concern worldwide, as they are transmitted through contaminated water or food, shed in high amounts within feces, and remain stable for a long time in the environment [220,221,222].

Accordingly, natural extracts of fruits, such as grapes and berries, have been tested extensively against enteric viruses to identify antiviral activities that may be exploited to develop new preventive or therapeutic agents, and thus to alleviate the burden of foodborne gastrointestinal viral diseases [223]. However, only for a few enteric viruses A-type PACs were observed to reproduce the antiviral activity of fruit extracts in which they have been characterized [191,224].

Rotavirus is a genus of non-enveloped, segmented double-stranded RNA viruses of the *Reoviridae* family. They are the major cause of acute gastroenteritis (AGE) in infants and young children worldwide, and the leading cause of viral diarrheal mortality with about 200,000 children under the age of 5 each year [225,226]. Even though implementation of rotavirus vaccination, as part of the routine childhood immunization program, proved to be effective in reducing AGE in countries where vaccines are used routinely, millions of children in several high-burden countries still lack access to rotavirus vaccine. Therefore, in the absence of effective control measures or treatment strategies, food extracts and juices endowed with anti-rotavirus activity may be of interest to control the infection and spread of AGE in those countries [223].

In this regard, in an early study, a cranberry juice was investigated for antiviral activity against the simian rotavirus SA-11 and found to protect monkey epithelial MA-104 cells from lytic infection. This antiviral effect was associated with the juice-mediated inhibition of the SA-11 hemagglutination activity, thus suggesting an interference of the juice’s components with adsorption of the rhesus rotavirus to the surface of host cells [192]. Subsequently, the same authors observed that dimeric and polymeric A-type PACs isolated from the cranberry extract indeed determined the loss of SA-11 viral capsid integrity in cell-free suspension, as measured by quantitative antigen capture assay of the virion VP6 protein [193]. Ultrastructural studies by transmission electron microscopy (TEM) then allowed visualization of a direct interaction of the A-type PACs with SA11 viral particles that were observed to be aggregated by the addition of PAC-As. It was therefore suggested that A-type PACs, by binding to and damaging rotavirus capsid proteins, affected the virus’ ability to attach to the host epithelial cell receptors, and in doing so they determined a reduction of viral infectivity [193].

Human noroviruses (HuNoVs) are non-enveloped single-stranded RNA positive viruses belonging to the *Caliciviridae* family [227]. HuNovs spread through the fecal-oral route and are the leading causative agent of AGE worldwide, with about 700 million cases and 200,000 deaths per year, and the second leading cause of AGE in children after rotavirus [228,229]. HuNoV infections therefore represent a major public health concern with considerable societal and economic outcomes. At present, however, no vaccines or antiviral agents have been licensed for prevention or treatment of HuNoV infections [230].

As for rotavirus, fruit extracts from different plants, such as cranberry, blueberry, pomegranate, and grape have been tested for anti-HuNoV activity. However, given the lack of a robust and reproducible cell system for in vitro HuNoV cultivation, surrogate caliciviruses, such as the feline calicivirus-F9 (FCV-F9) and the murine norovirus-1 (MNV-1) have been used extensively for investigating NoVs replication and pathogenesis, as well as in antiviral assays [231]. Using these animal caliciviruses, cranberry, blueberry and raspberry juices were observed to reduce infectivity of both FCV-F9 and MNV-1 as tested in virucidal assays [189,190,195,232,233]. As reported above, cranberries mainly contain A-type PAC, while blueberries contain mostly B-type PACs [145]; therefore, the two PAC types purified from the corresponding fruits were tested for anti-calicivirus activity, and found to reproduce the inhibitory activity of the corresponding juice, thus indicating that PACs characterized by both A-type and B-type linkages exerted antiviral activity against human enteric viral surrogates [189,190,233]. Especially, TEM analysis on FCV-9 particles exposed to PAC-As revealed major morphological alterations of capsid structure, thus suggesting the ability of cranberry’s type-A PACs to bind to the capsid proteins and altering virion structure in a manner such that the viral infectivity was compromised [190].

Taken together, the available data on the mechanism of action of A-type PACs against enteric viruses confirm the ability of these polyphenols to interact with proteins of the viral surface, thus causing alterations that, in turn, affect severely the virus’ ability to attach and/or enter into target cells. It is therefore tempting to envisage that PAC-As could potentially be exploited for the treatment and/or prevention of foodborne viral diseases.

### 3.5. Respiratory Viruses

The ongoing COVID-19 pandemic is proving that respiratory viral infections are a leading cause of morbidity and mortality worldwide, and a major societal and healthcare problem [234,235]. In fact, respiratory viruses replicate within the respiratory apparatus causing a broad range of respiratory tract infection (RTI) outcomes, ranging from asymptomatic to acute life-threatening diseases. These viruses spread through the respiratory secretions from an infected individual with three different mechanisms: direct/indirect contact, droplet spray, or aerosol (airborne transmission) [236]. RNA viruses are the predominant cause of RTIs in humans and include: influenza viruses (IV), parainfluenza viruses (PIV), metapneumoviruses (MPV), respiratory syncytial viruses (RSV), human rhinoviruses (hRV), enteroviruses, and human coronaviruses (hCoV). Among DNA viruses, adenoviruses (AdV), human bocavirus (hBoV), and reactivating herpesviruses in immunosuppressed individuals, can cause RTIs [234,235,236].

Although inhibitory activities against several of the above respiratory viruses have been described for many medicinal plant-derived extracts [237,238,239], PACs have been identified as the bioactive antiviral agents only in a few studies in which their inhibitory activity has been characterized against IVs and hCoVs.

Influenza remains a major public health challenge and, every year worldwide, IVs cause around 1 billion infections, 3–5 million of severe RTIs, and 290,000–650,000 respiratory deaths [240,241,242]. Even though seasonal vaccines represent the most effective measure for prevention and control of IV infections, antiviral agents are beneficial to reduce the burden of complications and case-fatality rates. However, the limited arsenal of anti-influenza drugs brings about challenges in the therapeutic management of influenza [241]. Therefore, new anti-influenza agents, effective against different IVs resulting from antigen variation, are urgently required and therefore intensely investigated.

To meet this need, PAC-enriched extracts derived from fruits and herbs have been examined in recent years as direct-acting anti-IV compounds [173,174,186,187,243,244]. Accordingly, oligomeric PAC-A and PAC-B were identified as the main antiviral principle of plant extracts. In a study, the dimeric proanthocyanidin epicatechin-3-*O*-gallate-(4b→8)-epicatechin-3′-*O*-gallate (procyanidin B2-di-gallate) was identified as the primary antiviral compound of an extract of garden sorrel (*Rumex acetosa*) able to inhibit influenza A viruses (IAV) H1N1, both laboratory strains and clinical isolates. Procyanidin B2-di-gallate was then proved to physically interact with the envelope hemagglutinin (HA) glycoprotein as alterations of electrophoretic mobility and immunoreactivity were observed [243]. The envelope of IAV contains two major glycoproteins, IAV hemagglutinin (HA) and neuraminidase (NA), that are essential for efficient infection and viral release from host cells. It was therefore suggested that PAC-B2 may interfere with the receptor binding pocket of HA and consequently affect the IV attachment to host cells. Specific penetration assays indeed confirmed that the PAC-B2 interfered with IAV entry [243]. In silico docking studies then indicated that procyanidin B2-di-gallate was predicted to interact with the receptor binding site of HA. In subsequent studies, PACs isolated from an extract of *Alpinia zerumbet*, an aromatic and medicinal plant, were observed to reduce the infectivity of the IAV H1N1 laboratory strain PR/8/34 in virucidal assays, thus indicating a direct interaction with viral particles [173]. The composition of *A. zerumbet*-derived PAC (AzPAC) was then determined and PAC-B2 and -B5 were identified as its major antiviral components. Interestingly, AzPAC was observed in quenching assays to strongly interact with recombinant HA and NA, and to affect the secondary structure of these viral glycoproteins in circular dichroism experiments [235]. It was therefore suggested that the impairment of IV replication caused by AzPAC was due to its direct interaction with IV envelope proteins in a manner that affected their function, thus preventing the attachment phase of the IV infection [174].

Regarding the anti-IV activity of A-type PACs, we have observed that a cranberry extract containing a high content of A-type dimers and trimers, potently inhibited the in vitro replication of both IAV and influenza B virus (IBV) [186]. Mechanistic studies revealed that this cranberry extract blocked the attachment and entry phases of IAV and IBV into target cells and exerted a virucidal activity against both IVs. These biological effects resulted from the ability of the extract to interact with the HA1 ectodomain of HA, as demonstrated by alteration of recombinant HA1 electrophoretic mobility with the occurrence of high molecular weight aggregates. Then, a detailed in silico docking simulation analysis indicated that among the different components of the chemical profile of the cranberry extract, PAC-A2 exhibited the best docking propensity to bind the HA protein with an affinity below 10 nM [186]. Subsequent docking simulation tests predicted the ability of PAC-A2 to bind first within the internal grooves of the HA structure by forming hydrogen bonds with phenylalanine and tryptophan residues, and then to other residues on the HA surface. These in silico predictions were then verified by fluorescence spectroscopy experiments that confirmed a direct interaction between the recombinant HA1 protein and purified PAC-A2. Importantly, purified AC-A2 was observed to potently inhibit both IAV and IVB replication with at about 5-log of reduction in viral titers, and to cause a complete loss of infectivity of IV particles in virucidal assays [186]. These antiviral assays therefore confirmed PAC-A2 as the major active anti-IV component of the cranberry extract. We concluded that the interactions of PAC-A2 with HA and the subsequent alterations in the viral protein function, determined the loss of infectivity of IV particles, thus preventing infection [186].

The disastrous consequence of COVID-19 indeed confirm that emerging coronaviruses are an indisputable major health threat, as proved by more than 600 million cases of COVID-19, including 6.5 million deaths worldwide [245]. In the last two decades, in fact, in addition to the endemic hCoVs (OC43, HKU1, 229E, and NL63), three highly-pathogenic hCoV, namely the severe acute respiratory syndrome coronavirus (SARS-CoV) in 2003, the Middle East respiratory coronavirus (MERS-CoV) in 2012, and SARS-CoV-2 in 2019 emerged as a result of zoonotic outbreaks [246]. These facts proved the urgent need for BSAAs that could be deployed against future hCoVs that could emerge in the future. For this purpose, plant extracts can be evaluated as sources of new anti-hCoV agents.

In an early study, Zhuang et al. [179] observed that a fraction of a cinnamon bark extract reduced the in vitro SARS-CoV replication so that the virus was exposed to the extract before cell infection. Among the chemicals purified from this fraction, PAC-A2 exerted the most potent inhibitory activity on SARS-CoV replication; thus, a virucidal effect was hypothesized as the event responsible for inhibition of infection [179].

With regard to SARS-CoV-2, a PAC-A-rich fraction from the leaves of rabbit-eye blueberry *(Vaccinium virgatum* Aiton) has been reported recently by Sugamoto et al. [194] to potently inhibit SARS-CoV-2 replication in Vero E6 cells (EC_50_ 1 µg/mL). For SARS-CoV-2, the maximum inhibitory effect was measured when the virus was treated with the PAC-A-rich fraction prior to infection, thus indicating that it could be administered as preventative treatment [194]. Interestingly, the PAC-A-rich fraction was observed to inhibit also the enzymatic activity of both the angiotensin-converting enzyme 2 (ACE2) receptor, which is the essential cell receptor for SARS-CoV-2, and the viral main protease chymotrypsin-like cysteine protease (3CL^pro^ or M^pro^), which is fundamental for processing viral polyproteins [246]. These findings suggest different inhibitory effects of the PAC-A-rich fraction against SARS-CoV-2. Indeed, the interference of PAC-As with viral particles and the cell surface receptor could result in the prevention of infection, while their direct-acting activity against M^pro^ may contribute to the overall antiviral activity after infection [194].

In addition to the thoroughly studied IV and hCoVs, the antiviral activity of PAC-As, in particular of the PAC-A2 dimer, was also observed for respiratory viruses of veterinary interest. In a first study, PAC-A2 purified from the bark of *Aesculus hippocastanum* was found to inhibit the in vitro replication of the Canine distemper virus (CDV), a *Morbillivirus* of the family of *Paramyxoviridae* that affects domestic and wild canines and other carnivores, and causes respiratory and systemic infections [247]. Time-of-addition experiments indicated the ability of PAC-A2 to exert its inhibitory activity during both early and late phases of the CDV replication cycle [248]. In a following investigation, lychee seeds-derived PAC-A2 was observed to exert a potent antiviral activity against the Porcine reproductive and respiratory syndrome virus (PRRSV) in alveolar macrophages, that represent the primary in vivo target cell type of PRRSV infection [249]. PRRSV is a single-stranded positive-sense RNA virus of the family of *Arteriviridae* and an endemic swine pathogen that causes pneumonia in piglets and growing pigs, thus determining one of the most economically costly diseases in the pig industry [250]. Although the specific mechanism of action against PRRSV was not detailed, PAC-A2 was monitored to prevent PRRSV replication by affecting both viral entry and progeny virus release. It was therefore suggested that PAC-A2 could be used to develop preventative and/or treatment interventions for PRRSV infections [250].

### 3.6. Non-Respiratory Emerging and Highly Pathogenic Viruses

Emerging viral infections represent a major concern for public health caused by both respiratory viruses and other zoonotic viral agents, as proved by the number of outbreaks and epidemics/pandemics occurring since the year 2000 [251,252]. Natural products have been tested against a number of such emerging viruses with the aim of developing control strategies; however, very limited information is available regarding the antiviral activity of PACs, and in particular PAC-A, as reported below.

One of the most deadly emerging viral diseases is the Ebola virus disease (EVD), caused by infection with Ebola virus (EBOV), an enveloped single-stranded RNA negative virus belonging to *Filoviridae* family [253]. EVD is characterized by hemorrhagic fever, shock from fluid loss and multi-organ failure with a high case fatality rate. However, there are no approved small molecules-based drugs for its effective treatment [254]. Several efforts have been advanced for the development of anti-EBOV agents targeting viral entry or viral genome replication [255,256]. The most advanced small molecule to treat EVD is remdesivir, a high cost broad-spectrum RdRp inhibitor available only for intravenous use, and thus difficult to use in the context of the low-income countries affected by EVD. Regarding natural products able to inhibit EBOV infection, a few reports have demonstrated the efficacy of PACs and their monomeric flavan-3-ols. In particular, the flavan-3-ol monomers gallic acid and epigallocatechin-3-gallate (EGCG) were reported to inhibit EBOV entry [257]. Time-of-addition assay in fact revealed that gallic acid likely interfered with the GP-mediated fusion in the late endosomes, while EGCG was found to inhibit the endoplasmic reticulum chaperone HSPA5, a host protein required for Ebola virus replication [257,258].

More recently, the screening of more than 500 extracts of medicinal plants collected in China allowed the identification of an anti-EBOV activity in a *Maesa perlarius* extract. Dimeric PAC and several flavan-3-ol monomers within this extract were found to be potent EBOV entry inhibitors at low micromolar concentrations. By docking analysis and microscale thermophoresis technology, the authors determined that these compounds exhibited virucidal potency by interacting with EBOV glycoprotein, and the most efficient antiviral compound was the PAC-B2 [259].

Dengue fever is caused by a flavivirus and represents one of the major public health concerns affecting almost 400 million people worldwide; it is endemic in at least 100 countries in the tropics and subtropics [260] The Dengue virus (DENV) is transmitted through the bite of female *Aedes aegypti* or *Aedes albopictus* mosquitoes. Human infection can range from asymptomatic cases to a severe disease characterized by severe plasma leakage leading to shock, bleeding or organ impairment [260]. Neither vaccines nor specific antivirals are available. Plant extracts were proposed as source of antivirals to treat Dengue fever [261]. In particular, Kimmel and co-workers [261] evaluated the antiviral effect of oligomeric PACs derived from unripe apple peels (rich in PAC-B) using cultured human PBMC derived from healthy subjects. Addition of purified oligomeric PACs (trimers and tetramers), immediately after the infection, reduced viral titer of 1.5 log. It was also observed that these PACs directly interacted with DENV particles, thus reducing virus infectivity. Finally, the authors reported that unripe apple peels-derived PACs also modulated the innate immune response in infected PBMCs, likely contributing to the overall inhibition of DENV replication in target cells [261].

Mayaro virus (MAYV) is an emerging mosquito-borne alphavirus (*Togaviridae*) affecting individuals in permanent contact with forested areas in tropical South America. This enveloped virus with single-stranded, positive-sense RNA genome causes nonspecific febrile illness and long-lasting arthritis/arthralgia [262]. MAYV diffusion is increasing and is a potential candidate to cause large-scale epidemics; therefore, the design and development of candidates for anti-MAYN viral drugs are urgently needed [263]. To this end, many strategies have been applied to identify antiviral molecules, including the use of plant extracts [263]. In particular, PACs obtained from methanol extraction of *Maytenus imbricata* (*Celastraceae*) roots showed a concentration-dependent virucidal effect on MAYV. This compound acted directly in MAYV particles and not on host cells as their treatment before infection did not show any antiviral effect. Interestingly, experiments with dialyzed virus suggested an irreversible inhibition of viral infectivity upon PACs treatment, thus suggesting a strong interaction between PACs and viral envelope or physical damage of the virion [264].

Crimean-Congo hemorrhagic fever virus (CCHFV) is an enveloped single-stranded negative sense RNA virus with a tri-segmented genome belonging to the *Nairoviridae* family. CCHFV causes an emerging tick-borne viral disease widely distributed across Africa, Southern Europe, the Middle East and Asia. Human infections can present as a spectrum from the absence of symptoms through mild signs, to severe hemorrhagic illness with a fatality rate up to 30% [265]; there is no FDA-approved vaccine or specific antiviral [266]. In 2018, CCHF was included in the WHO Blueprint list of priority diseases to promote the research for vaccines and drugs [267]. Nevertheless, to date very few papers reporting the discovery of anti-CCHF agents have been published [268]. In this regard, we have recently reported that a cranberry (*V. macrocarpon* Aiton) extract rich in PAC-As inhibits CCHFV infection [185]. To investigate the antiviral mechanism of this cranberry extract, we used the Hazara virus, a nairovirus model of CCHFV that can be handled in Biosafety Level (BSL)-2 Laboratories, instead of BSL-4 required for CCHFV. Time-of-addition experiments showed that the cranberry extract inhibited viral infection by targeting early stages of the replication cycle. In particular, specific viral attachment assays indicated that the main antiviral mechanism is the inhibition of virus attachment to target cells, thus suggesting interactions between bioactive PAC-As and Hazara virus glycoproteins. This hypothesis was further supported by the observation of a virucidal activity of the extract when incubated with HAZV particles before the infection of cells [185].

Overall, PACs, and in some cases PAC-As, have been reported to inhibit non-respiratory emerging viruses by affecting primarily the early step of viral replication cycles, likely as a consequence of alterations of the functions of viral proteins required for attachment and/or entry into host cells.

## 4. Biological Activities of PAC-As Other than the Antiviral Effects

In many cases, the antiviral action of PAC-A has been associated also with other important properties of these polyphenols, such as the antioxidant, antibacterial, antidiabetic, antihypoglycemic, cardioprotectant, and immunomodulatory activities. Therefore, the following paragraphs will summarize these properties as a compendium of the biological activity of PAC-A.

### 4.1. Antioxidant Activity

In general, the antioxidant activity of a PAC-A-containing plant extract increases with increasing degrees of A-type PAC polymerization [269], as we noticed in our PCA analysis (Figure 2). A-type PACs may reduce oxidative stress by acting as free radical scavengers, and by affecting signaling pathways associated with cellular oxidative stress homeostasis [18]. In the Malvaceae family, a radical-scavenging effect was shown in *Adansonia digitata* pericarp (fruit wall) fruits, which contained an A-type PAC trimer [31], whereas in cocoa (*Theobroma cacao*) epicatechin-containing dimers showed a strong antioxidant power [124]. In the Sapindaceae family, *Aesculus turbinata* polyphenol polymers with doubly linked A-type interflavans linkages exhibit potent antioxidant activities [32], while in *Litchi chinensis* A-type dimers and trimers qualify the fruit stones and the pericarp of this plant as a raw material for polyphenol extracts exerting significant antioxidant properties [78,79,80,81]. In the same family, *Dimocarpus longan* contains PAC trimers-octamers that show promising antioxidant activities which could be applied as potential functional food components [62]. In the Fabaceae family, peanut (*Arachis hypogaea*) skin A-type PACs were effective against H_2_O_2_-induced oxidative stress damage in prostate cancer DU145 cells [35], and thus they have been proposed as an inexpensive source of antioxidants for use as functional ingredients in foods or dietary supplements [42]. In the same family, *Spatholobus suberectus* fractions enriched in PAC monomers and oligomers exerted antioxidant activity in MCF-7 breast cancer cells [120]. In the *Rosaceae*, apple (*Malus domestica*) extracts showed a high antioxidant potential using 2,2-diphenyl-1-picrylhidrazyl (DPPH) and oxygen radical absorbance capacity (ORAC) methods, whereas in plum (*Prunus domestica*) the antioxidant activity was even higher [86]. In the Ericaceae, cranberry (*Vaccinium macrocarpon*) radical scavenging and antioxidant activities were attributable to their composition of PACs [151], while in *Gaultheria procumbens* the leaf antioxidant activity was found to change according to the harvesting season [66,67]. Finally, in grapevine (*Vitis vinifera*, Vitaceae) glial cultures pretreated with grape seed-derived type-A PACs showed improved viability after H_2_O_2_-induced oxidative stress [160].

### 4.2. Antibacterial Activity

Cranberry (*Vaccinium macrocarpon*, Ericaceae)-derived PACs are unique in their structure with a higher percentage of A-type bonds, compared with PACs from other commonly consumed fruits [270]. It is well known that cranberry extracts and juices have an anti-bacterial effect and are thus traditionally used to treat cystitis and UTIs [15,137,138,139,140,141,147], such as those caused by uropathogenic *Escherichia coli* [271]. In the same plant genus, *V. myrtillus*-derived A2-type PACs contained in juices were effective against bacterial strains of *Asaia lannensis* and *Asaia bogorensis* [117]. The same effects were found also in *Ribes nigrum* (*Grossulariaceae* family) [117]. *A. hypogaea* skin extracts display anti-microbial activity due to its A-type PAC content, able to prevent pathogen infection [36]. A-type PACs of *Cinnamomum zeylanicum* (Lauraceae family) were effective against uropathogenic *E. coli* multidrug-resistant strains and showed a marked antibiofilm activity [50]. Adhesion of *Streptococcus pyogenes* to human airway epithelial (HEp-2) cells was found to be inhibited by A-type PACs contained in *Pelargonium sidoides* (Geraniaceae family) extracts [97], whereas *Pinus pinaster* (Pinaceae family) bark extracts containing A-type PAC dimers showed bactericidal actions against *Staphylococcus aureus* and *E. coli* [102].

### 4.3. Antidiabetic and Hypoglycemic Activity

A recent meta-analysis revealed that there is a significant effect of PAC supplementation on blood glucose levels and, once in the liver, PACs oligomers may modulate hepatocyte functions and interfere with glucose uptake and metabolism [19]. In the family Lauraceae, the genus *Cinnamomum* contains A-type PACs that exerted hypoglycemic effects. In *C. cassia,* the main A-type PAC oligomers could reverse palmitic acid-induced dysfunction of glucose-stimulated insulin secretion in primary cultured islets, improved the insulin concentration in the blood and pancreas, and (as *C. japonica*) improved insulin sensitivity in type 2 diabetes mellitus [46,47,48]. A-type PAC oligomers of C. *tamala* improved the insulin concentration in the blood and pancreas [47], whereas *C. zeylanicum* A-type PACs potentiated insulin action, and may be beneficial in the control of glucose intolerance and diabetes [56,57]. In the Ericaceae family, the A-type doubly linked PAC trimers of *V. corymbosum* and *V. myrtillus* acted as antidiabetic substances [127], whereas in *Areca catechu* (Arecaceae family) the presence of A-type PACs ameliorates the streptozocin-induced hyperglycemia by regulating gluconeogenesis [43]. Excellent inhibitory effects on α-glucosidase were found in extracts of *Pyracantha fortuneana* (Rosaceae family), and these effects were due to the alteration of the active site catalytic configuration of the enzyme in such a manner as to reduce substrate binding affinity [110].

### 4.4. Lipid Lowering Effects and Cardiovascular Protection

PACs can interfere with lipid metabolism affecting intestinal absorption of lipids [272] and liver secretion of chylomicrons and lipoproteins [19]. For instance, PAC-A2 significantly reduces cellular lipid accumulation and restricts ox-LDL-induced cellular oxidative stress and inflammation [273]. Grape seed (*V. vinifera*, *Vitaceae*)-derived PACs can regulate lipid metabolism and significantly decreased the expression of pro-inflammatory cytokines, thus exerting hypolipidemic and potential anti-inflammatory effects in the liver [164]. The consumption of grape seed PACs has been related to lower oxidized low-density lipoprotein particles and LDL cholesterol [161,163], to improve dyslipidemia associated with a high-fat diet, mainly by repressing lipogenesis and VLDL assembly in the liver [162]. Furthermore, grape seed PACs exerted a pronounced effect on the cholesterol and triglyceride levels [165] and, by inhibiting oxidation of LDL, showed an antiatherosclerotic activity [166]. On the other hand, A-type PACs of peanut (*A. hypogaea*) skin extracts exert protection against hepatic steatosis induced in rats fed with a high-fat diet by inhibiting the absorption of dietary lipid and chylomicron secretion by enterocytes [34]. Litchi (*L. chinensis*, *Sapindaceae*) pericarp-derived extracts rich in PAC-As have cardioprotection effects on myocardial ischemia injury and lower serum malondialdehyde contents in high-fat/cholesterol-dietary hamsters [74,75]. *Mandevilla moricandiana* (*Apocynaceae* family) A-type PAC trimers have been observed to induce a concentration-dependent vasodilation on aortic rings through the NO pathway, with the involvement of histamine H1 and estrogen ER alpha receptors [87]. Finally, A-type PACs of cranberry (*V. macrocarpon*) have been reported to inhibit platelet aggregation and adhesion, to inhibit enzymes involved in lipid and lipoprotein metabolism, to induce endothelium-dependent vasorelaxation, and to increase reverse cholesterol transport and decrease total and LDL cholesterol [152].

### 4.5. Immunomodulatory Activity

Proanthocyanidin rich foods can markedly influence the immune responses to enteric infections. Mechanistic studies have demonstrated that dietary PACs exert direct modulatory effects on immune cell signaling, by boosting the recruitment of immune cells and suppressing the amount of pro-inflammatory cytokines. Some anti-inflammatory effects of PAC stem from a direct modulation of mucosal immune cells [274]. The prebiotic effect of PAC has been speculated to be primarily responsible for their anti-inflammatory and immunomodulatory activity [275,276]. Some pathologies, like psoriasis, involve inflammatory mechanisms that interact with immune homeostasis and prevent autoimmune diseases by suppressing immune responses [167]. Grape seed (*V. vinifera*) extracts containing PACs have been reported to act on the immune system by regulating the differentiation of inflammatory T cells and possess the ability of multidirectional regulation of immunity by maintaining the dynamic balance of immunity in psoriasis [168,277]. Dietary *V. vinifera*-derived PACs promote the DNA repair-dependent stimulation of the immune system following the functional activation of dendritic cells and effector T cells [169,170,171,172], whereas *V. macrocarpon* PACs improve immune function and modify cytokine and signal transduction pathways [153]. Grape seed PACs were found to attenuate TNF-alpha and IL-1 beta-induced IL-6 production, and decreased IL-17-stimulated ERK 1/2, p38, and JNK MAPK activities in A549 human pulmonary epithelial cells [278]. Grape seed extracts were also found to inhibit the NF-kappa B pathway in human prostate carcinoma DU145 cells [279]. Finally, transcription of inflammatory factors such as myeloperoxidase, interleukin (IL)-1 beta, IL-6, and tumor necrosis factor alpha (TNF-alpha) was also down-regulated in lung tissue by grape seed PACs [280]. Overall, these data indicate a potential immunomodulatory effect of PACs.

## 5. Conclusions

The devastating consequence of COVID-19 is indisputable evidence of the need for BSAAs effective also against viruses that may emerge from future zoonoses. Indeed, the availability of an antiviral arsenal that includes such BSAAs would make it possible to immediately protect human populations from an emerging viral disease, while waiting for the development of the new virus specific vaccines and DAAs.

The exploitation of natural products to derive BSAAs can meet this urgent need. However, some critical issues must be faced, such as the production of highly active and standardized extracts, the identification of the bioactive components responsible for the antiviral activity, and the characterization of the mechanism(s) of action, which is often related to a synergistic cooperation among different components. The studies we have reviewed here suggest that A-type PACs can overcome these hurdles, and highlight facts that sustain the feasibility of PAC-As as BSAAs candidates.

Firstly, the main mechanism of the antiviral action of PAC-As appears to be the same for most of the viruses examined regardless of whether they are non-enveloped or enveloped DNA or RNA viruses, thus making PAC-As suitable for interventions against new or hitherto unrecognized viruses. Indeed, when examined in detail, the BSAA activity of PAC-As has proven to originate from the inhibition of the virus attachment to the surface of target cells. In many studies, this anti-adhesive effect of PAC-As has been associated with their ability to interact directly with those virion capsid proteins or glycoproteins that are essential for attachment and entry, thus preventing access to their normal binding partners on target cells. This general mechanism of the antiviral activity of PAC-As could result from the natural propensity of polyphenols to bind and aggregate proteins [281,282]. In this regard, it has been proposed that different types of chemical interactions, such as hydrogen bonding, van der Waals and electrostatic interactions, or even covalent linkages may contribute to the formation of protein-polyphenol complexes [282]. The endurance of electrophoretic mobility alterations induced by PAC-A dimers and trimers on HSV gD and gB, as well as on IV HA1, upon boiling of protein samples in SDS sample buffer, sustains the view that the exposure of purified viral glycoproteins to PAC-As results in the formation of covalent linkages between PAC-As molecules and viral proteins [186,188]. These covalent interactions may result in protein-protein crosslinking, as most PACs have two or more reactive quinone moieties [283]; this would explain the smearing and disappearance of glycoprotein bands that we and others have observed in electrophoretic mobility shift assay experiments with purified viral glycoproteins [186,188,204,215,243].

Taken as a whole, the PAC-As-protein interactions may lead to alterations of viral capsid or envelope protein structures and functions, or to masking/blocking their binding sites to cellular receptors, eventually resulting in the inhibition of binding of the viral particles to cell receptors (Figure 3). Accordingly, this mechanism of action advocates the potential application of PAC-As-containing agents as BSAAs in the treatment or prevention of current viral infections, as well as in the preparedness for future emerging viral threats.

Secondly, given the virucidal activity of PAC-As, it is possible to envisage that PAC-As-containing formulations would allow inactivation of a broad range of infecting viruses and therefore prevention of many viral diseases. For example, considering the significant global incidence, morbidity, and mortality rates of both viral RTIs and sexually transmitted infections (STIs), the development of new, safe, attachment/entry inhibitors based on PAC-A-containing agents could provide a realistic method of antiviral intervention, as well as blocking virus shedding and transmission by close personal contact. Regarding RTIs, local application of formulations rich in PAC-As in the upper respiratory tract, administered as tablets or chewing gums or through inhaling devices, would allow the inactivation of infecting virus and thus prevention of infection [283]. Likewise, PAC-As-containing topical microbicides to be applied directly to the genital tract would prevent the establishment of a viral STI, such as HSV or HIV-1 [284,285]. Furthermore, the topical use of PAC-As-based formulations (as aerosolized suspensions or creams) would overcome limitations that might occur due to unsatisfactory PAC-As levels in the blood following systemic treatments. However, such formulations must satisfy the two fundamental requirements of efficacy and safety. Regarding the toxicity of PAC-As, we have recently observed that when tested for effects on the viability of human cells, different purified PAC-As were found to be safe with noteworthy low cytotoxicity values [286]. Moreover, from this perspective, the widespread use of different formulations of dried cranberry extracts, naturally rich in PAC-As [146], for the prevention of urinary tract infections (UTIs) sustains the high safety profile of PAC-As-containing products to develop broad-spectrum antiviral agents of natural origin suitable to prevent infections [287,288]. Interestingly, the antibacterial activity of PAC-As could be valuable also for the prevention of bacterial superinfections of the respiratory tract that may follow viral RTIs (due to both virus- and immune-mediated damage of the respiratory mucosa).

Thirdly, in many low- and middle-income countries, antiviral drugs are often beyond the reach of the people who need them most or are unavailable. Therefore, for those health systems a reliable, affordable, and high-quality supply of low cost antivirals could be essential to control viral infections [289]. Economic models indicate that in developing countries manufacturers of pharmaceuticals and phytopharmaceuticals should be able to charge substantially lower prices with respect to industrialized countries, without impairing their profits and with no reduction of the therapeutic power. Although in middle-income and developing countries prices are already substantially discounted, compared to developed countries, an economic foundation for fair antiviral drug pricing could be based on widely available BSAAs with low costs of production [290]. Low-cost production PAC-A-based BSAAs could be therefore of particular interest to those low-income countries where viral RTIs or sexually transmitted diseases still have a high incidence. In this perspective, the development of PAC-A-rich phytocomplexes as new BSAAs, could not only be advantageous from an economic point of view compared to more expensive purification procedures or chemical synthesis of specific PAC-A molecules, but could allow better exploitation of the synergistic and holistic effects of different bioactive PACs naturally present in a plant-derived extract [19,27]. Thus, a PAC-A-enhanced phytocomplex could be the most suitable candidate for both preclinical and clinical development of PAC-A-based BSAA, owing to the presence of the most active components that contribute to the overall antiviral activity of the plant extract.

Taken together, these considerations support the suitability of A-type PACs to constitute the antiviral active agent of plant-derived formulations for the development of effective BSAAs, that can be rapidly deployable against current viral infections and future emerging viruses.

## Figures and Tables

**Figure 1 molecules-27-08353-f001:**
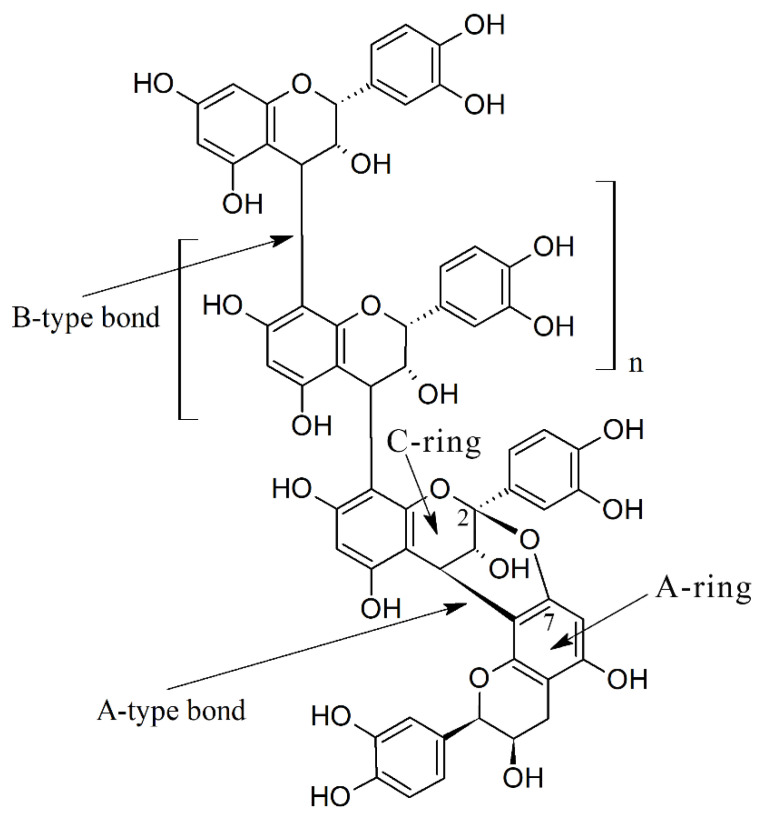
Structure of proanthocyanidins showing A-type and B-type interflavanic bonds and the position of A and C rings along with the numbers of carbons involved in the C–O bounds.

**Figure 2 molecules-27-08353-f002:**
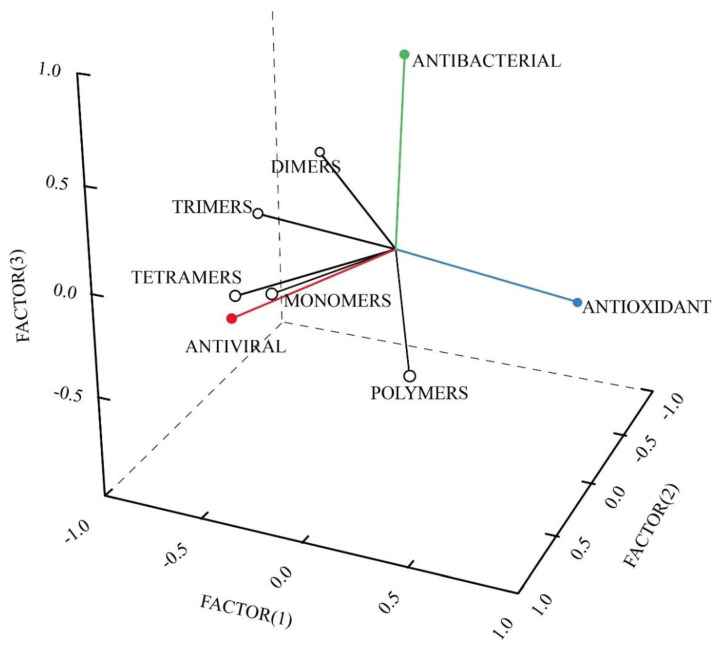
Factor loading plot from the Principal Component Analysis (PCA) performed on data summarized in Table 1 and Table 2 by considering the different degrees of PAC-A polymerization and the three main biological effects. Antiviral activity is associated mainly with A-type PAC monomers, dimers, trimers and tetramers. Antioxidant activity is correlated with A-type PAC polymers with a degree of polymerization > 5. Antibacterial activity is correlated primarily to PAC-A dimers and trimers. Varimax rotation; total variance explained by the three factors: 26% factor (1), 24% factor (2) and 17% factor (3).

**Figure 3 molecules-27-08353-f003:**
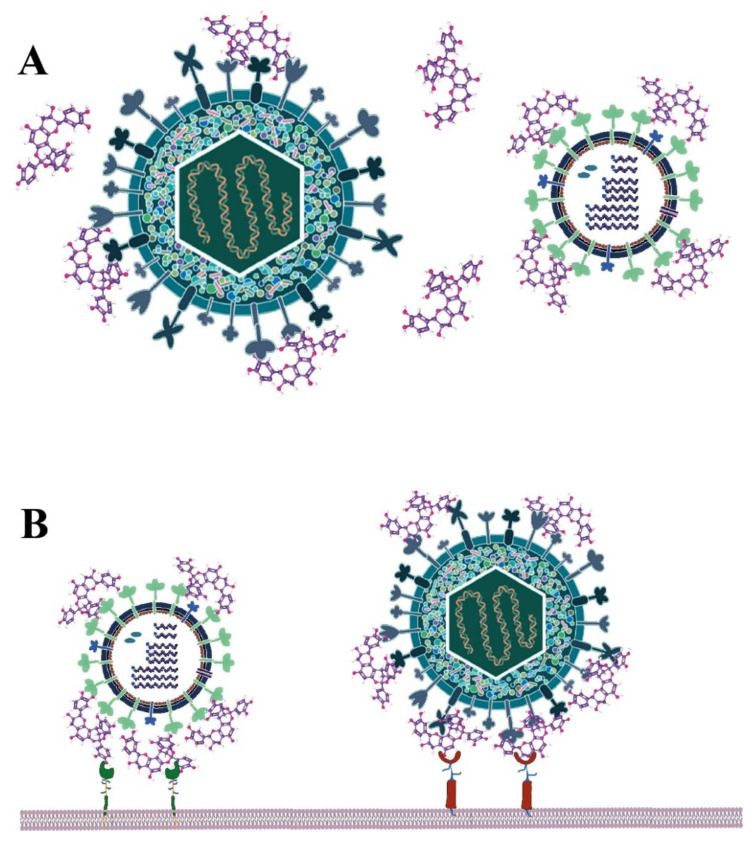
A proposed mechanism of action for the broad-spectrum antiviral activity of type-A PACs. (**A**) Tampering with virion capsid or envelope glycoproteins. (**B**) Masking/blocking virion binding sites to cellular receptors.

**Table 1 molecules-27-08353-t001:** Natural sources of A-type PACs and biological properties.

Natural Source	A-Type PACs	Properties	References
*Adansonia digitata*	dimers	antioxidant	[31]
*Aesculus turbinata*	procyanidins *	antioxidant	[32]
*Aglaonema commutatum* var. *maculatum*	trimers	chemical composition	[33]
*Aglaonema crispum*	dimers	chemical composition	[33]
*Arachis hypogaea*	dimers, trimers	cardiovascular diseases, dyslipidemia antioxidant prevention of pathogen infection chemical composition anti-inflammatory antioxidant	[34] [35] [36] [21,37,38,39,40] [41] [42]
*Areca catechu*	dimer	hypoglycemic	[43]
*Calluna vulgaris*	dimers, trimers	chemical composition	[44]
*Cinnamomum cassia*	dimers, oligomers	oxidative conversion of B- to A-procyanidins antidiabetic	[45] [46,47,48]
*Cinnamomum japonica*	oligomers	hypoglycemic	[48]
*Cinnamomum tamala*	oligomers	antidiabetic	[47,49]
*Cinnamomum zeylanicum*	trimers, tetramers	multidrug resistance, biofilm inhibitory activity antiasthmatic, antiallergic anti-inflammatory antiallergic attenuation of the reduction in glutamate uptake anti- vascular endothelial growth factor (VEGF) antidiabetic prevention of neurodegeneration	[50] [51] [52] [53] [54] [55] [56,57] [58]
*Coffea arabica*	trimers	chemical composition	[59]
*Crataegus pinnatifida* var. *major*	procyanidins	antioxidant	[60]
*Dimocarpus longan*	dimer	health-beneficial bioactivity	[61]
	procyanidins	antioxidant	[62]
*Ecdysanthera utilis*	monomers, dimers	immunomodulator	[63]
*Ephedra equisetina*	procyanidins	chemical composition	[64]
*Ephedra intermedia*	procyanidins	chemical composition	[64]
*Ephedra sinica*	procyanidins	chemical composition	[64,65]
*Gaultheria procumbens*	trimers	antioxidant	[66,67]
*Ixora coccinea*	dimers	antioxidant, antibacterial	[68]
*Laurus nobilis*	trimers	antioxidant	[69]
*Litchi chinensis*	dimers, trimers	chemical composition antioxidant cardioprotection alteration of oligomers in the gastrointestinal system bioavailability antioxidant absorption and urinary excretion bacterial bioconversions	[70,71,72] [73] [74,75] [76] [77] [78,79,80,81] [82] [83,84]
*Lotus americanus*	procyanidins	chemical composition	[85]
*Malus domestica*	dimers	antioxidant	[86]
*Mandevilla moricandiana*	trimers	antioxidant	[87]
Microbiota (faecal and gut)	procyanidins	Inability to cleave A-type linkages preventing of biofilm formation antioxidant degradation by human gut microbiota	[88,89] [90] [91] [92]
*Paullinia cupana*	trimers	Anti-inflammatory, antioxidant	[93,94]
*Paullinia pinnata*	trimers, tetramers	antihelminthic	[95,96]
*Pelargonium sidoides*	trimers	antiadhesive	[97]
*Persea americana*	dimers, trimers, tetramers, procyanidins	chemical composition	[98,99]
*Pheonix dactylifera*	dimers	chemical composition	[100]
*Pinus massoniana*	trimers, tetramers	increased modulus of elasticity of dentin	[101]
*Pinus pinaster*	dimers	antioxidant, bactericidal	[102]
*Polygonum cuspidatum*	dimers	chemical composition	[103,104]
*Prunus domestica*	dimers	chemical composition antioxidant	[99,105] [86]
*Prunus dulcis*	procyanidins	chemical composition	[106,107]
*Prunus spinosa*	dimers, trimers	chemical composition	[44,108]
*Pteris vittata*	procyanidins	antioxidant	[109]
*Pyracantha fortuneana*	procyanidins	antidiabetic	[110]
*Pyrus pyrifolia*	trimers	chemical composition	[111]
*Rhizophora apiculata*	monomers	chemical composition	[112]
*Rhizophora mangle*	procyanidins	chemical composition	[113]
*Rhododendron ferrugineum*	trimers	vitality and the proliferation rates of epithelial HaCaT keratinocytes	[114]
*Rhododendron formosanum*	trimers	induction of autophagy antioxidant	[115] [116]
*Ribes nigrum*	dimers	bacterial growth and cell adhesion	[117]
*Rubus idaeus*	procyanidins	chemical composition	[118]
*Rumex obtusifolius*	trimers	chemical composition	[119]
*Spatholobus suberectus*	procyanidins	antioxidants, inhibitor of breast cancer	[120]
*Tectaria macrodonta*	trimers	chemical composition	[33]
*Theobroma cacao*	procyanidins	chemical composition absorption antioxidant	[37,121,122] [123] [124]
*Vaccinium ashei*	dimers, dodecamers	chemical composition	[125]
*Vaccinium consanguineum,*	monomers, dimers, trimers, tetramers, procyanidins	chemical composition	[126]
*Vaccinium corymbosum*	trimers	antidiabetic	[127]
*Vaccinium floribundum*	monomers, dimers, trimers, tetramers, procyanidins	chemical composition	[126]
*Vaccinium macrocarpon*	monomers, dimers, trimers, tetramers, procyanidins	chemical composition urinary tract infections (UTIs) antiaging bioavailability transported across Caco-2 cells antioxidant cardiovascular health immune system	[30,128,129,130,131,132,133,134,135,136] [14,15,23,137,138,139,140,141,142,143,144,145,146,147] [148] [149] [150] [151] [152] [153]
*Vaccinium myrtillus*	dimers, trimers	antidiabetic chemical composition bacterial growth and cell adhesion	[127] [44,154] [117]
*Vaccinium oxycoccus*	monomers, dimers, trimers, tetramers, procyanidins	chemical composition	[135]
*Vaccinium poasanum*	monomers, dimers, trimers, tetramers, procyanidins	chemical composition	[126]
*Vaccinium vitis-idaea*	dimers, trimers	chemical composition	[44,135]
*Vicia faba*	dimers	chemical composition	[155]
*Vitis vinifera*	dimers, trimers, tetramers, procyanidins	chemical composition inhibition of alpha-glucosidase promotion of DNA repair in dendritic cells in UVB-exposed skin. decreases the progression of airway inflammation antioxidant control of lipid metabolism immune system	[156] [157] [158] [159] [160] [161,162,163,164,165,166] [167,168,169,170,171,172]

* The term procyanidins indicates A-type PACs with degrees of polymerization (DP) 5 < DP < 12.

**Table 2 molecules-27-08353-t002:** Antiviral activity of plants-derived PAC-As.

Natural Source	A-Type PACs	Virus and Mechanism of Action	References
*Alpinia zerumbet*	procyanidins *	influenza A virus, inhibition of attachment, virucidal	[173,174]
*Chamaecrista nictitans*	procyanidins	HSV-1 and HSV-2, NA	[175]
*Cinnamomum cassia*	dimers, oligomers	HIV-1, interaction with envelope glycoproteins	[176]
*Cinnamomum zeylanicum*	trimers, tetramers	HIV-1, inhibition of attachment HCV, inhibition of attachment SARS-CoV, virucidal	[177] [178] [179]
*Ixora coccinea*	trimers	HIV-1, inhibition of Vpu activity; HCV, NA	[180]
*Litchi chinensis*	dimers, trimers	HSV-1 and Coxsackie virus B3, NA	[181]
*Pinus maritima*	procyanidins	HIV-1, inhibition of entry and replication HCV, inhibition of replication	[182] [183]
*Pomelia pinnata*	dimers	HIV-1, inhibition of integrase activity	[184]
*Sambucus nigra*	dimers	HIV-1, interaction with envelope glycoproteins	[176]
*Theobroma cacao*	dimers	HSV and HIV, NA	[124]
*Vaccinium macrocarpon*	monomers, dimers, trimers, tetramers, procyanidins	nairovirus, inhibition of attachment	[185]
		influenza A and B virus, inhibition of attachment	[186,187]
		and entry, virucidal	
		HSV-1 and HSV-2, inhibition of entry	[188]
		human norovirus surrogates: murine norovirus (MNV-1), feline calicivirus (FCV-F9), virucidal	[189,190]
		reovirus, NA	[191]
		rotavirus, inhibition of attachment, interaction with capsid proteins	[192,193]
*Vaccinium myrtillus*	dimers, trimers	SARS-CoV-2, inhibition of entry and replication	[194]
		HA, NA	[195]
		HCV, inhibition of replication	[196]
*Vitis vinifera*	dimers, trimers, tetramers, procyanidins	rotavirus, affecting virion integrity HIV-1, inhibition of entry by down-modulation of co-receptors	[193] [197]

* The term procyanidins indicates A-type PACs with degrees of polymerization (DP) 5 < DP < 12; NA: not available.

## Data Availability

Not applicable.

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
