# Peer review of "Tackling the Future Pandemics: Broad-Spectrum Antiviral Agents (BSAAs) Based on A-Type Proanthocyanidins"

_molecules, 2022, doi:10.3390/molecules27238353_

Round 1

Reviewer 1 Report

In this review, the authors summarize the current knowledge about the in vitro antiviral properties of a specific class of plant-derived polyphenols, and their potential to be developed as broad-spectrum antivirals that can be quickly used in case of a novel epidemic with a known or unknown virus. They explain their chemical properties as well as their biological properties, before concentrating on their antiviral activity against a broad range of human viruses.  

The manuscript is generally very well and clearly written, and fills a scientific gap as until now the antiviral properties of A-type PACs have not been summarized in literature.

General comments

Section 3: According to this reviewer, the biological activities other than antiviral are less relevant for the scope of this manuscript. This section should be shortened and moved to the end of the manuscript, or even better discarded and included in Table 1, which can be expanded somewhat to include specific examples or other relevant information.

Table 2: maybe it would be better to merge both tables, as the way it is represented now, is not very well structured. For example, Cinnamomum zeylanicum has anti-HCV activity according to Table 1, but this is not included in Table 2, and almost all organisms with antiviral properties are also represented in table 1 but this is only clear after thorough back and forth reading. It could for example be an option to highlight the ones with antiviral properties by bold or underlined.

Is it possible to include graphical representations of the antiviral mechanism of action of the compounds against specific viruses? This would further highlight the common mechanism of action against a broad range of viruses and also improve reading attractiveness.

Could the authors include an additional section on mechanism of action of these compounds? Also, there is too little attention for the possibility of resistance selection against these compounds, as this is very well plausible, e.g. by changes in envelope protein structures or alternative entry routes. This could be further discussed in more detail.

In the concluding remarks, possible limitations of further development should be addressed, and perhaps the most suitable candidate for further (pre)clinical development could be proposed?

Specific small comments

Line 35-45: two very long sentences

Line 46: remove ‘in’

Line 120: add point

Line 132: ‘method’ remove s

Line 135-136: ‘to identify’?

Line 139: ‘represented’

Table 1: what is meant with ‘chemical composition’ as biological property?

Figure 2: somewhat unclear to interpret, maybe add lines to improve 3D effect?

Reviewer 2 Report

The authors extensively reviewed all activities regarding the A-Type Proanthocyanidins. My suggestions are as follows;
1. Table 1 should indicate the purity of the PAC-A and the level of activity.  

2. Table 2 should include antiviral activities such as anti-attachment, anti-replication, or any specific target protein.

3. The biological activities like lipid-lowering properties seems unrelated. However, certain lipid-lowering drugs are reported as antivirals. It might be a good transition for explaining why the authors reviewed these properties. 

Reviewer 3 Report

The review article “Tackling the Future Pandemics: Broad-Spectrum Antiviral Agents (BSAAs) Based on A-Type Proanthocyanidins” by Maffei, Salata and Gribaudo deals mainly with the antiviral effects of A-type proanthocyanidins (PAC-As). These plant-derived polyphenols can form oligo- or polymers that exert protective effects against pathogens, insects and herbivores. Beneath their antioxidant, antibacterial and immunomodulatory effects, PAC-As show broad-spectrum antiviral activity by interfering with viral entry mechanisms.

After a well written introduction about the pros and cons of Direct-Acting Antivirals versus Host-Targeting Antivirals, the authors provide a short outline about the chemistry and the biological properties of A-type PACs and from which plants they can be isolated. Afterwards the biological activities of PAC-A-containing plant extracts regarding their therapeutic impact as antioxidant, antibacterial, antidiabetic, antihypoglycemic, cardioprotectant and immunomodulatory compounds were shortly summarized.  

The main focus of the manuscript is on broad-spectrum antiviral activity of A-type PACs in a large set of different viruses and virus families like HSV, HIV, HCV, hepatitis E virus, several respiratory viruses and other highly pathogenic viruses like EBOV, DENV or Crimean-Congo hemorrhagic fever virus (CCHFV).

Antiviral activity of A-type PACs is mainly correlated to monomers, dimers, trimers and tetramers via the inhibition of viral attachment to the surface of host cells and subsequent entry into the cell. The EC50 values are in the micromolar range with a Selectivity Index (SI) ranging from 3 to 212 (a PAC-B from blueberry leaves that inhibit HCV replication showed the highest SI). However, in most cases SI is below 100 (in the range between 10 to 30).

PAC-A-containing fractions can interact with envelope glycoproteins of different viruses e.g. IND02 (contain A-type PAC trimers and pentamers) bind to gp120 of HIV-1 types that use CXCR4 or CCR5 as co-receptors. It was also shown that Procyanidin B2-di-gallate physically interact with the envelope hemagglutinin (HA) glycoprotein of IAV H1N1. In general, PACs interact with proteins of the viral surface, thus attachment and/or entry of viruses into target cells is aggravated. However, in case of SARS-CoV-2 the activity of the viral main protease 3CLpro seems also to be inhibited by a PAC-A-rich fraction from the leaves of rabbit-eye blueberry.

In my opinion the manuscript is in a suitable form for publication in the Journal “Molecules” after some minor revision.

The authors should critically discuss some important issues regarding toxicity (see low SI), synthesis of PAC-As under GMP conditions (this is an important issue in drug development) and efficiency of the PACs regarding their antiviral activity in more detail. What is the log-phase reduction of virus titers for the different viruses? For example, the authors mentioned that DENV titers can be reduced about one log in human PBMCs. Other natural compounds with broad-spectrum antiviral activity can reduce virus titers about 2-4 log phases!

I was also wondering if the effects of PACs are potent enough to prevent viral infections simply by the inhibition of virus entry via an unspecific aggregation of surface proteins? Should there be other drugs used in combination with PACs that inhibit viral replication in cells? How long is the duration of the antiviral effect of PACs in cellular infection systems? What about the propensity of polyphenols to bind and aggregate proteins? This seems to be a drawback that could be responsible for their toxicity. However, a more critical discussion is needed to improve the manuscript.

Other minor points:

Lane 394: the Authors shown that….   Check English spelling

Lane 426: During during….  Please delete "during"

Lane 435: Please correct  ….PACS inhibited HBV infection both in cell lines than in primary human hepatocytes….  into    ….and in primary human hepatocytes….

Lane 703: change viral title …   into viral titer
